# Dp71 and intellectual disability in Indonesian patients with Duchenne muscular dystrophy

**Kristy Iskandar**[1]*, **Agung Triono**[1], **Sunartini**[1], **Ery Kus Dwianingsih**[2], **Braghmandita Widya Indraswari**[1], **Ignatia Rosalia Kirana**[1], **Gabriele Ivana**[1], **Retno Sutomo**[1], **Suryono Yudha Patria**[1], **Elisabeth Siti Herini**[1], **Gunadi**[3]

1 Department of Child Health, Faculty of Medicine, Public Health, and Nursing, Universitas Gadjah Mada, Yogyakarta, Indonesia, 2 Department of Anatomical Pathology, Faculty of Medicine, Public Health, and Nursing, Universitas Gadjah Mada, Yogyakarta, Indonesia, 3 Pediatric Surgery Division, Department of Surgery/Genetics Working Group, Faculty of Medicine, Public Health, and Nursing, Universitas Gadjah Mada/ Dr. Sardjito Hospital, Yogyakarta, Indonesia

* kristy.iskandar@ugm.ac.id

## Abstract

### Introductions

Duchenne muscular dystrophy (DMD) is an X-linked recessive progressive muscular disease marked by developmental delays due to mutations in the *DMD* gene, which encodes dystrophin. Brain comorbidity adds to the burden of limited mobility and significantly impacts patients' quality of life and their family. The changes of expression of dystrophin isoforms in the brain due to *DMD* gene mutations are thought to be related to the cognitive and neurobehavior profiles of DMD.

### Objectives

This cross-sectional study aimed to characterize cognitive and neurodevelopmental profiles of patients with DMD and to explore underlying genotype-phenotype associations.

### Methods

Patients with DMD aged 5–18 years from Dr Sardjito Hospital and Universitas Gadjah Mada Academic Hospital from 2017–2022 were included. Multiplex ligation-dependent probe amplification and whole exome sequencing were used to determine mutations in the *DMD* genes. Cognitive function was measured by intelligence quotient testing using the Wechsler Intelligence Scale for Children and adaptive function tests with Vineland Adaptive Behavior Scales. The Autism Mental Status Exam and Abbreviated Conner's Rating Scale were used to screen for autism spectrum disorder (ASD) and attention deficit and hyperactivity disorder (ADHD), respectively.

### Results

The mean total IQ score of DMD patients was lower than that of the general population ($80.6 \pm 22.0$ vs $100 \pm 15$), with intellectual disability observed in 15 boys (29.4%). Of the 51 patients with DMD, the Dp71 group had the lowest cognitive performance with a total IQ

**Funding:** This research is funded by Faculty of Medicine, Public Health and Nursing, Universitas Gadjah Mada, Indonesia and 3 billion, Inc, South Korea. The funders had no role in study design, data collection and analysis, decision to publish, or preparation of the manuscript.

**Competing interests:** The authors have declared that no competing interests exist.

score (46 ± 24.8; $p = 0.003$), while the Dp427 group and Dp140 group's total IQ scores were 83.0 ± 24.6 and 84.2 ± 17.5 respectively. There were no DMD patients with ASD, while 4 boys (7.8%) had comorbidity with ADHD.

## Conclusion

Boys with DMD are at higher risk of intellectual disability. The risk appears to increase with mutations at the 3' end of the gene (Dp71 disruption). Moreover, Dp71 disruption might not be associated with ADHD and ASD in patients with DMD.

## Introduction

Duchenne muscular dystrophy (DMD) is a severe neuromuscular disorder that affects skeletal muscles in 1:5,000 births [1, 2]. DMD is an X-linked recessive progressive muscular disease due to mutations in the *DMD* gene, which causes the absence or damage of dystrophin [2, 3]. Dystrophin serves as a link between the internal cytoskeleton and extracellular matrix proteins, which stabilize muscle contractions. In patients with DMD, the absence of dystrophin results in muscle damage due to impaired membrane integration and cellular necrosis [3].

Dystrophin has an essential role in brain development and function. Furthermore, dystrophin-associated glycoprotein complexes are involved in ion channels and postsynaptic membrane receptors during synaptogenesis. Dystrophin protein (Dp) variants are named based on their weight in kilodaltons [4]. The intact dystrophin protein (Dp427) and other shorter isoforms (Dp260, Dp140, Dp116, Dp71) are expressed specifically in muscle, heart, and brain tissues [5].

Previous studies have shown that the distribution of dystrophin in the amygdala, hippocampus and cerebral cortex is associated with cognitive phenotypes [5]. The incidence of cognitive decline was higher in the deletions of exons 45–52, affecting the Dp140 and Dp71 isoforms, suggesting the importance of these isoforms in cognition and brain function [6, 7]. Mutations that cause changes in the dystrophin isoform Dp140 (exons 45–55) were found in 5 of 8 (62.5%) DMD patients with autism spectrum disorder (ASD), whereas mutations occurring between exons 31 and 62 had stereotypical symptoms of ASD [8].

The prevalence of ASD, attention deficit and hyperactivity disorder (ADHD), epilepsy and obsessive-compulsive disorder were higher in boys with dystrophinopathies, including DMD and Becker muscular dystrophy (BMD) compared to normal populations. Learning and behaviour problems in DMD can occur with or without cognitive impairment and can be detected early in development. Intellectual disability is common in children with DMD (30%), while in the general population, the prevalence is only 1% [6]. ADHD is the most common neurobehavioral comorbidity in DMD. As many as 30–50% of children with DMD also have ADHD [7], while the incidence of ASD in people with DMD is 4–37%. The average intelligence quotient (IQ) of people with DMD, which is 85, is lower than the population mean, while the IQ considered normal in the general population is 100±15 [8].

## Materials and methods

### Participants

This cross-sectional study was conducted on patients aged 5–18 years old who were diagnosed with DMD. Diagnoses were confirmed through a DMD gene mutation test from January, 2017

to June, 2022 at Dr. Sardjito General Hospital and Universitas Gadjah Mada Academic Hospital, Yogyakarta, Indonesia [9]. Multiplex ligation-dependent probe amplification (MLPA) was used to determine exon deletions or duplications in the *DMD* gene. If MLPA showed no deletions or duplications, point mutations were discovered using whole-exome sequencing (WES) at 3billion, Inc, Seoul, South Korea. Patients who are not able to perform cognitive and neurobehavior assessments were excluded.

Patients older than 12 years old and/or their parents or guardians (for patients <12 years old) signed a written informed consent form to be included in this study. The Institutional Review Board of the Faculty of Medicine, Public Health and Nursing, Universitas Gadjah Mada/Dr. Sardjito General Hospital approved this study (KE/FK/0894/EC/2020 and KE/FK/0180/EC/2021).

## Data collection and variables

Baseline characteristics of the patient were collected using questionnaires. Cognitive function was measured by IQ score using the Wechsler Intelligence Scale for Children (WISC). The Vineland Adaptive Behavior Scales (VABS) were used to evaluate standardized adaptive behaviour measurements. Further screening on ASD and ADHD was conducted using Autism Mental Status Exam (AMSE) and Abbreviated Conner's Rating Scale (ACRS), respectively. Cognitive function, ASD and ADHD were assessed based on the Diagnostic and Statistical Manual of Mental Disorders-5th edition (DSM-5) criteria. We divided patients into three groups based on the mutations in the *DMD* gene: (1) Dp427 group (mutation before the 31st exon), (2) Dp140 group (mutation between the 31st to 62nd exons), and (3) Dp71 group (mutation after 63rd exon).

## Statistical analysis

Frequency was presented as percentages. Normality of the data was checked using the Shapiro-Wilk test, with *p*-values >0.05 considered as normally distributed data. Data were presented as mean ± standard deviation (SD) for normally distributed data and median (minimum-maximum) for nonnormally distributed data. The differences between mutation groups were analyzed using Kruskal-Wallis tests and Fisher Exact tests for categorical data, while Independent T-test, one-way ANOVA and Kruskal-Wallis tests were used to analyze numeric data, with *p*-values <0.05 considered as statistically significant. Data entry and analysis were performed using SPSS v.25 (IBM Corp., Armonk, NY).

## Results

Data were obtained from 62 patients diagnosed with DMD based on genetic mutation analysis at Dr. Sardjito General Hospital and Universitas Gadjah Mada Academic Hospital, Yogyakarta, Indonesia from January 2017 to June 2022. Eight patients were excluded due to no cognitive or neurobehavior data available, while 3 patients were excluded due to BMD phenotype. Out of 51 patients with DMD who underwent genetic testing, exon deletions were found in 36 (70.6%) patients, exon duplications were found in 11 (21.6%) patients, a point mutation was found in 4 (7.8%) patients, and intron mutation was found in 1 (1.9%) patient (Fig 1).

The patients were divided into three groups based on dystrophin isoform disruption: 10 patients were in the Dp427 group, 37 patients were in the Dp140 group, and only four patients were in the Dp71 group. The detail of the mutation identified in all subjects was presented in Table 1.

There was no significant correlation between dystrophin disruption groups and the patients' characteristics (Table 2).

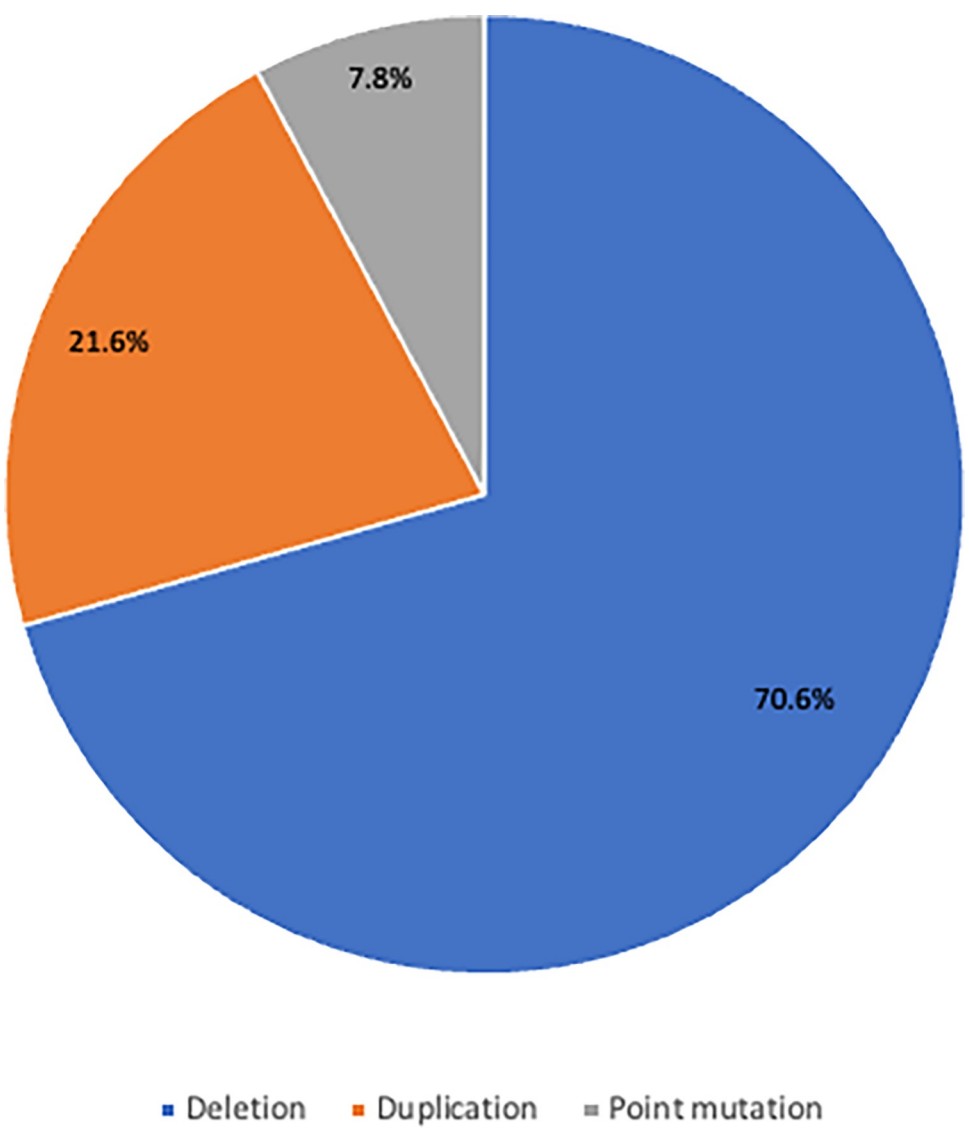

**Fig 1. Distribution of types of mutation in patients with DMD.**

The average IQ test results for all patients were 80.6 ± 22.0 for total IQ, 82.9 ± 23.6 for verbal abilities, and 75.8 ± 19.7 for performance abilities. The Dp71 group significantly had the lowest cognitive function with total IQ score of 46.0 ± 24.8 compared to the other two groups ($p = 0.003$) accompanied with scores for verbal abilities of 49.75 ± 26.73 ($p = 0.009$), and for performance abilities of 45.3 ± 24.8 ($p = 0.002$). There were no significant differences of adaptive function profiles in the three groups (Table 3).

The incidence of intellectual disability differed significantly between the Dp71 group and the other two groups ($p = 0.009$). All patients within the Dp71 group had intellectual disabilities. No significant findings of ADHD nor any patients with ASD were found among the patients with DMD in our study (Table 4).

No significant association was found between IQ test results and prevalence of intellectual disability, ADHD, and ASD with mutation type in each dystrophin disruption group. However, there was a significant difference in adaptive function, especially in ABC and communication in the Dp427 group (Table 5).

**Table 1. Mutation identified.**

| Name | Age | Mutation |
|---|---|---|
| **Dp427 (N = 10)** | | |
| MFR | 13 | Deletion of exons 8–30 |
| AAZ | 15 | Deletion of exons 3–17 |
| KN | 8 | Deletion of exon 1 |
| LR | 13 | Deletion of exons 5–7 |
| MT | 12 | Deletion of exons 1–4 |
| D | 13 | Deletion of exons 15 |
| APS | 8 | Deletion of exons 2–15 |
| KM | 10 | Deletion of exons 8–29 |
| RP | 8 | Duplication of exons 10–17 |
| **Dp140 (N = 37)** | | |
| NRR | 9 | Deletion of exon 45 |
| ARP | 8 | Deletion of exons 46–50 |
| PBT | 13 | Deletion of exons 7–43 |
| RWN | 9 | Deletion of exons 46–50 |
| AP | 11 | Deletion of exon 51 |
| BWC | 13 | Deletion of exons 49–50 |
| MHH | 16 | Deletion of exons 53–54 |
| NPP | 11 | Deletion of exons 46–51 |
| SY | 8 | Deletion of exons 46–50 |
| AM | 13 | Deletion of exons 48–50 |
| FS | 17 | Deletion of exons 60 |
| MDZ | 10 | Deletion of exon 52 |
| YSF | 15 | Deletion of exons 46–48 |
| GAP | 12 | Deletion of exon 51 |
| RGM | 12 | Deletion of exons 42–52 |
| NSA | 17 | Deletion of exons 17–43 |
| AR | 13 | Deletion of exons 3–44 |
| DNM | 12 | Deletion of exons 51–54 |
| MKH | 7 | Deletion of exons 53–55 |
| ZRA | 10 | Deletion of exons 45–50 |
| MA | 10 | Deletion of exon 52 |
| AN | 10 | Deletion of exons 3–44 |
| MNA | 8 | Deletion of exons 47–50 |
| E | 8 | Deletion of exons 44–50 |
| R | 4 | Deletion of exons 44–50 |
| RA | 9 | Deletion of exon 44 |
| MNH | 11 | Duplication of exons 2–44 |
| AJ | 8 | Duplication of exons 44–51 |
| MNA | 12 | Duplication of exons 44–48 |
| IAP | 13 | Duplication of exons 51–62 |
| MFH | 14 | Duplication of exon 40 |
| RD | 7 | Duplication of exons 12–44 |
| RA | 17 | Duplication of exons 2–44 |
| EM | 12 | NM 004006.2:c.6614+2T>C |
| IA | 9 | NM_004006.3:c.8086del |
| AS | 10 | NM_004006.3:c.9224+5G>A |

*(Continued)*

**Table 1.** (Continued)

| Name | Age | Mutation |
|---|---|---|
| HMS | 14 | NM_004006.3:c.5363C>A |
| **Dp71 (N = 4)** | | |
| NF | 10 | Deletion of exons 56–74 |
| RR | 12 | Duplication of exons 63 |
| GF | 9 | Duplication of exons 53–63 |
| MYS | 7 | Duplication of exons 53–63 |

**Table 2.** Patients' characteristics.

| Characteristics | Dystrophin disruption groups | | | p value |
|---|---|---|---|---|
| | **Dp427 (N = 10)** | **Dp140 (N = 37)** | **Dp71 (N = 4)** | |
| **Age** | 10.9 ± 2.6 | 11.1 ± 3.1 | 9.5 ± 2.08 | 0.572 |
| **Disease onset** | 6.7 ± 2.9 | 5.8 ± 2.3 | 5.8 ±4.3 | 0.639 |
| **Ambulatory** | | | | 0.349 |
| Early ambulatory | 4 (36%) | 4 (10.3%) | 0 (0%) | |
| Late ambulatory | 0 (0%) | 2 (5.1%) | 1 (25%) | |
| Non-ambulatory | 5 (45%) | 24 (61.5%) | 2 (50%) | |
| **Family history of disease** | 2 (20%) | 10 (27%) | 1 (25%) | 0.923 |
| **Physical Examination** | | | | |
| Pseudohypertrophy | 4 (36%) | 19 (51.4%) | 1 (35%) | 0.420 |
| Gower sign | 5 (45%) | 18 (48.6%) | 3 (75%) | 0.759 |
| **Laboratories result** | | | | |
| CK (U/L) | 4314 (1,077–17,238) | 6,788 (444–16,550) | 6,681 (3,335–7,875) | 0.604 |
| CKMB (U/L) | 145 (91–399) | 288 (49–831) | 222 (173–311) | 0.384 |
| Positive biopsy | 3 (27%) | 11 (29.7%) | 3 (75%) | 0.365 |

Note: significance set as $p<0.05$; CK, creatine kinase; CKMB, creatine kinase myocardial band.

**Table 3.** Association between cognitive profile and dystrophin disruption group.

| | Dystrophin disruption group | | | p value |
|---|---|---|---|---|
| | **Dp427** | **Dp140** | **Dp71** | |
| | **(N = 10)** | **(N = 37)** | **(N = 4)** | |
| **WISC** | | | | |
| Total IQ Score | 83.0 ± 24.6 | 84.2 ± 17.5 | 46 ± 24.8 | 0.003* |
| Verbal | 85.7 ± 28.4 | 87.0 ± 17.5 | 49.8 ± 26.7 | 0.009* |
| Performance | 82.6 ± 18.6 | 78.2 ± 15.3 | 45.3 ± 24.8 | 0.002* |
| **VABS** | | | | |
| ABC | 47.5 ± 22.3 | 46.6 ± 15.4 | 40.67 ± 26.35 | 0.850 |
| Communication | 43 (33–84) | 59 (20–91) | 40 (19–67) | 0.683 |
| Daily Living | 69.0 ± 20.6 | 42.6 ± 23.1 | 39.3 ± 34.4 | 0.063 |
| Socialization | 65.5 (45–86) | 61 (20–99) | 53 (19–87) | 0.491 |

Note: significance set as $p<0.05$; ABC, Adaptive Behavior Composite; WISC, Wechsler Intelligence Scale for Children; VABS, Vineland Adaptive Behavior Scales.

**Table 4. Prevalence of intellectual disability, ADHD, and ASD based in dystrophin disruption groups.**

|  | Dystrophin disruption groups | | | Total N = 51 | p value |
|---|---|---|---|---|---|
|  | Dp427 (n,%) | Dp140 (n,%) | Dp71 (n,%) |  |  |
| Intellectual disability | 3 (30%) | 8 (21.6%) | 4 (100%) | 15 (29.4%) | 0.009* |
| ADHD | 1 (10%) | 3 (8.1%) | 0 (0) | 4 (7.8%) | 0.819 |
| ASD | 0 (0) | 0 (0) | 0 (0) | 0 (0) | - |

Note: significance set as $p<0.05$; ADHD, attention deficit disorder; ASD, autism spectrum disorder.

Overall, no significant association was found between the severity of symptoms with IQ test results and adaptive function profiles and the prevalence of intellectual disability, ADHD, and ASD in each dystrophin disruption group (Table 6)

## Discussion

Exon deletions were found in 36 out of 55 patients (70.6%), higher than the result from a previous study in Indonesia, in which DMD patients had 44.1% deletions detected by multiplex PCR [10]. This result was similar to a study conducted by Aartsma-Rus *et al.*, which found ~68% deletion. We found 21.6% for duplications and 7.8% for point mutations, in which one patient (1.9%) had intron mutation. A previous study reported duplications and point mutations were ~11% and ~20% of patients, respectively [11].

Dystrophin loss is associated with anatomic and physiological adaptations that contribute to the cognitive deficits present in patients with DMD. Dystrophin is one of the proteins that build up the Dystrophin-Glycoprotein Complex (DGC). DGCs play a role in the organization of the gamma-aminobutyric acid (GABA) receptors on neurons and the aquaporin-4 protein complex (AQP4) on glial cells. Accordingly, defects in DGC processing and formation are

**Table 5. Association between cognitive profile and prevalence of intellectual disability, ADHD, and ASD with mutation type in each dystrophin disruption groups.**

|  | Dp427 (N = 10) | | | | Dp140 (N = 37) | | | | Dp71 (N = 4) | | | |
|---|---|---|---|---|---|---|---|---|---|---|---|---|
|  | Deletion (N = 9) | Duplication (N = 1) | Point Mutation (N = 0) | p value | Deletion (N = 26) | Duplication (N = 7) | Point Mutation (N = 4) | p value | Deletion (N = 1) | Duplication (N = 3) | Point Mutation (N = 0) | p value |
| **WISC** | | | | | | | | | | | | |
| Total IQ Score | 81.1 ± 25.6 | 98.00 | N/A | 0.553 | 82.3 ± 17.7 | 86.3 ± 19.0 | 93.7 ± 13.6 | 0.547 | 37.0 | 49.0 ± 29.5 | N/A | 0.758 |
| Verbal | 83.4 ± 29.5 | 104.00 | N/A | 0.531 | 84.3 ± 17.1 | 91.8 ± 8.5 | 97.3 ± 22.0 | 0.425 | 35.0 | 54.7 ± 30.4 | N/A | 0.632 |
| Performance | 81.4 ± 19.5 | 92.00 | N/A | 0.624 | 75.7 ± 17.0 | 81.0 ± 5.2 | 90.0 ± 3.6 | 0.314 | 35.0 | 48.7 ± 29.2 | N/A | 0.724 |
| **VABS** | | | | | | | | | | | | |
| ABC | 36.6 ± 9.8 | 76.00 | N/A | 0.022* | 45.5 ± 15.5 | 56.0 ± 2.8 | 53.5 ± 24.4 | 0.584 | 33.0 | 43.0 ± 26.9 | N/A | 0.778 |
| Communication | 43.4 ± 10.0 | 84.00 | N/A | 0.021* | 52.5 ± 18.5 | 64.5 ± 14.9 | 57.5 ± 25.7 | 0.702 | 37.0 | 43.0 ± 24.0 | N/A | 0.849 |
| Daily Living | 58.2 ± 23.0 | 70.00 | N/A | 0.664 | 42.3 ± 23.2 | 48.5 ± 14.9 | 57.0 ± 38.3 | 0.623 | 20.0 | 42.3 ± 33.7 | N/A | 0.624 |
| Socialization | 61.8 ± 13.4 | 86.00 | N/A | 0.176 | 59.5 ± 21.3 | 61.0 ± 1.4 | 58.0 ±26.2 | 0.986 | 51.0 | 53.7 ± 34.0 | N/A | 0.952 |
| **Prevalence of intellectual disability, ADHD, and ASD** | | | | | | | | | | | | |
| Intellectual disability | 3 (33.3%) | 0 | 0 | 1.000 | 5 (19.2%) | 1 (14.3%) | 1 (25%) | 0.892 | 1 (100%) | 3 (100%) | 0 | - |
| ADHD | 1 (11.1%) | 0 | 0 | 1.000 | 2 (7.7%) | 1 (14.3%) | 0 | 0.705 | 0 | 0 | 0 | - |
| ASD | 0 | 0 | 0 | - | 0 | 0 | 0 | - | 0 | 0 | 0 | - |

Note: significance set as $p<0.05$; ABC, Adaptive Behavior Composite; WISC, Wechsler Intelligence Scale for Children; VABS, Vineland Adaptive Behavior Scales; ADHD, attention deficit disorder; ASD, autism spectrum disorder.

**Table 6. Association between cognitive profile and prevalence of the intellectual disorder, ADHD, and ASD with ambulatory status in each dystrophin disruption groups.**

| | Dp427 (N = 10) | | | Dp140 (N = 37) | | | Dp71 (N = 4) | | |
|---|---|---|---|---|---|---|---|---|---|
| | Ambulatory (N = 5) | Non ambulatory (N = 5) | P value | Ambulatory (N = 13) | Non ambulatory (N = 24) | P value | Ambulatory (N = 2) | Non ambulatory (N = 2) | P value |
| **WISC** | | | | | | | | | |
| Total IQ Score | 96.0 ± 32.2 | 72.6 ± 11.2 | 0.168 | 85.0 ± 17.7 | 83.7 ± 17.8 | 0.847 | 52.5 ± 21.9 | 39.5 ± 34.7 | 0.698 |
| Verbal | 104.3 ± 34.2 | 70.8 ± 10.8 | 0.075 | 91.6 ± 20.2 | 84.5 ± 15.9 | 0.339 | 56.0 ± 29.7 | 42.5 ± 33.2 | 0.730 |
| Performance | 87.0 ± 24.7 | 79.0 ± 14.2 | 0.558 | 85.8 ± 13.9 | 74.2 ± 14.9 | 0.065 | 49.5 ± 20.5 | 41.0 ± 36.8 | 0.802 |
| **VABS** | | | | | | | | | |
| ABC | 53.5 ± 31.8 | 38.0 ± 10.8 | 0.387 | 56.1 ± 9.2 | 43.8 ± 18.7 | 0.121 | 52.5 ± 27.6 | 28.5 ± 12.4 | 0.384 |
| Communication | 63.5 ± 26.2 | 42.5 ± 11.4 | 0.180 | 61.1 ± 13.4 | 51.2 ± 21.4 | 0.285 | 52.0 ± 21.2 | 31.0 ± 17.0 | 0.388 |
| Daily Living | 76.0 ± 8.5 | 52.3 ± 21.7 | 0.228 | 59.4 ± 23.0 | 38.3 ± 24.5 | 0.080 | 50.5 ± 43.1 | 23.0 ± 5.7 | 0.466 |
| Socialization | 69.5 ± 23.3 | 64.0 ± 14.5 | 0.729 | 68.6 ± 9.4 | 54.6 ± 23.8 | 0.189 | 69.0 ± 25.5 | 37.0 ± 25.5 | 0.336 |
| **Prevalence of intellectual disability, ADHD, and ASD** | | | | | | | | | |
| Intellectual disability | 1 (16.7%) | 2 (40%) | 1.000 | 2 (15.4%) | 5 (20.8%) | 1.000 | 2 (100%) | 2 (100%) | - |
| ADHD | 0 | 1 (20%) | 1.000 | 1 (7.7%) | 2 (8.3%) | 1.000 | 0 | 0 | - |
| ASD | 0 | 0 | - | 0 | 0 | - | 0 | 0 | - |

Note: significance set as $p < 0.05$; ABC, Adaptive Behavior Composite; WISC, Wechsler Intelligence Scale for Children; VABS, Vineland Adaptive Behavior Scales; ADHD, attention deficit disorder; ASD, autism spectrum disorder.

associated with brain abnormalities across a wide spectrum, ranging from mild cognitive impairment to brain migration disorders, apart from muscle disorders [12]. There is evidence of impaired central nervous system (CNS) architecture, dendritic abnormalities, and neuronal loss in patients with DMD. Electroencephalogram (EEG) abnormalities, brain atrophy on computerized tomography (CT) scan and magnetic resonance imagery (MRI) in patients with DMD have also been reported previously [13, 14].

Assessment of patients' intellectual ability was done by measuring cognitive domains using the WISC test and the adaptive scale using the VABS test. The average IQ score in the patients with DMD in this study was 80.6 ± 22.0. This is lower than the average IQ score in the general population, which was 100 ± 15. This result is in accordance with research results in a multi-center meta-analysis of patients with DMD in Europe, which previously reported the average IQ score of 81.3 ± 5.1 [5, 13]. This is also in line with study results from Asian populations in Japan and India [15, 16]. Studies in various populations show that the mean IQ score in patients with DMD is 1–1.5 SD below the normal mean [7, 17]. In the present study, verbal IQ was higher than performance in all three dystrophin isoform disruption groups. This finding is in contrast to a previous study which found verbal IQ was lower than performance IQ in patients with DMD. They suggested the impairment in verbal abilities was caused by weaknesses in sequential auditory and visual information processing that reduced both attention and memory [13].

In this research, intellectual disability was found in 15 children (29.4%). Previous studies reported the frequency of intellectual disability appeared in 18.6–27% of patients with DMD in the Caucasian race [6, 12]. Cognitive function in the Dp71 group was significantly lower ($p < 0.05$), both in total IQ scores and verbal and performance abilities. The total IQ score in the Dp71 group corresponds to intellectual disability, which is <70. Ricotti et al. reported that there was an increased risk of cognitive impairment in boys with mutations at the 3' end of the

*DMD* gene. This *DMD* gene mutation, which is located downstream of exon 63, encodes an isoform of the dystrophin protein called Dp71, which is the most expressed dystrophin isoform in the brain [4, 18].

The adaptive function of patients with DMD in this study was measured by VABS. The mean of adaptive function in children with DMD was also low in this study, which was 46.2 ± 17.4. There was no significant correlation between dystrophin disruption groups and adaptive function. Nevertheless, patients with DMD were reported to have a significant delay in adaptive function profiles compared with unaffected family members. Adaptive function profiles in patients with DMD were 1 SD lower than controls, with the most significant differences observed in communication and motoric skills [17].

In this study, there were no patients with suspected ASD based on the results of screening using the Autism Mental Status Exam. This measuring instrument is used for children over five years, with a specificity of 90.5% and a sensitivity of 81.2% [19]. Other different diagnostic methods that are more sensible and more specific could have been used (i.e., ADI-ADOS-2). ASD is reported to occur in 4–37% of patients with DMD. However, the measurement tools for screening and diagnosis of ASD vary between reported studies. Simone *et al.* reported that 19% of patients with DMD met the ASD criteria using the Autism Diagnostic Interview-Revised tool [20]. In comparison, Ricotti *et al.* reported a figure of 21% using the 3D diagnostic interview, which is a high sensitivity test tool considered to be more specific and reliable by taking into account the possibility of overestimation [15].

Dystrophin has an essential role in brain development. Lack of dystrophin and its mutations disrupt brain isoforms and are more likely to be associated with neurodevelopmental disorders [18]. The exact mechanism between ASD and DMD remains unclear. The previous study found that mutations in the distal *dystrophin* gene, which is associated with the production of short isoforms such as Dp71, are more likely to be associated with ASD compared to mutations of upstream parts of the gene [21]. However, this contradicts the findings of a previous study, which found that upstream gene mutations are also associated with the severity of autistic disorders [15]. This might be due to the loss of shorter isoforms progressively affecting the dystrophin as a whole [20].

There were four patients (7.8%) diagnosed with ADHD based on the DSM-5 diagnostic criteria according to the screening result score using the Abbreviated Conner Rating Scale 13. Another instrument for diagnosing ADHD is available, for example, Nepsy II in its dedicated subparts. There was no significant relationship between the location of the *DMD* gene mutation with the incidence of ADHD in this study. In a cohort study, ADHD was confirmed in 32%, compared to ADHD in general populations, which was 3–7% [22]. Previously, one study reported a range between 12–50% [7]. Another study by Ricotti *et al.* reported that 44% of patients with DMD presented with attention deficit disorder [15]. ASD and ADHD were found to be frequently associated with mutations affecting Dp140 and dystrophin short isoforms, including Dp71, between exons 62 and 63. Mutations on the middle and 3' end of the gene may have a higher risk of manifesting attention disorders in children, where 25% of DMD patients with ADHD have mutations of the upstream exon 44 [7]. Another theory proposed that steroid therapy may cause behavioral problems in children. Long-term use of corticosteroid therapy may result in memory and attention problems [23]. However, this result is still controversial [7].

Correlation between types of mutation (deletion, duplication, and point mutation) and severity of symptoms (ambulatory and non-ambulatory) with cognitive profiles and prevalence of intellectual disability and ADHD in each dystrophin disruption group were also analyzed. No significant difference was found in the IQ mean and the prevalence of intellectual disability and ADHD in each type of mutation. At the same time, there is a significant

correlation between types of mutation and adaptive and communication functions in the Dp427 group. Previous studies also reported no significant difference in mean IQ in each type of mutation [24]. Furthermore, the significant difference in cognitive profiles, adaptive profiles, and the prevalence of intellectual disability and ADHD was not observed for the severity of symptoms in each group.

This study is a cross-sectional study conducted with a valid assessment to measure neurodevelopment in male children with DMD with a broad spectrum of mutations, using genetic testing diagnostic techniques according to international guidelines over five years. This is the first study ever reported in Indonesia which examined the neurobehavior profiles of patients with DMD concerning the genotypes of the *DMD* gene mutations. As a single-center study, limits exist in its ability to depict the condition of the entire Indonesian population.

## Conclusions

Boys with DMD are at higher risk of intellectual disability. The risk appears to increase with mutations at the 3' end of the gene (Dp71 disruption). However, dystrophin disruption might not be associated with ADHD and ASD in patients with DMD. Further multicenter studies with larger sample populations are needed to confirm our findings.

## Supporting information

**S1 Data.**
(XLSX)

## Acknowledgments

We are sincerely grateful to our patients and patients' families who were willing to participate in this research. We would also like to thank Dwi Susilawati, MA, S.Psi., Psychology, Melina Dian Kusumadewi, S.Psi., M.A., Psychology, and all the staff and nurses who helped in caring for the patients' wellbeing.

## Author Contributions

**Conceptualization:** Kristy Iskandar, Agung Triono, Braghmandita Widya Indraswari, Elisabeth Siti Herini, Gunadi.

**Data curation:** Kristy Iskandar, Ery Kus Dwianingsih, Gunadi.

**Formal analysis:** Kristy Iskandar, Braghmandita Widya Indraswari, Suryono Yudha Patria.

**Investigation:** Kristy Iskandar, Braghmandita Widya Indraswari, Retno Sutomo, Elisabeth Siti Herini, Gunadi.

**Methodology:** Kristy Iskandar, Agung Triono, Braghmandita Widya Indraswari, Elisabeth Siti Herini.

**Project administration:** Kristy Iskandar, Suryono Yudha Patria.

**Resources:** Gunadi.

**Software:** Kristy Iskandar.

**Supervision:** Sunartini, Retno Sutomo, Suryono Yudha Patria, Elisabeth Siti Herini.

**Validation:** Agung Triono, Suryono Yudha Patria, Elisabeth Siti Herini, Gunadi.

**Visualization:** Suryono Yudha Patria.

**Writing – original draft:** Kristy Iskandar, Ignatia Rosalia Kirana, Gunadi.

**Writing – review & editing:** Kristy Iskandar, Gabriele Ivana, Retno Sutomo, Suryono Yudha Patria, Gunadi.

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
