## [Decision Letter · Decision Letter 0]

12 May 2022

PONE-D-22-04415Dp71 and intellectual disability in Indonesian patients with Duchenne muscular dystrophyPLOS ONE

Dear Dr. Iskandar,

Thank you for submitting your manuscript to PLOS ONE. After careful consideration, we feel that it has merit but does not fully meet PLOS ONE’s publication criteria as it currently stands. Therefore, we invite you to submit a revised version of the manuscript that addresses the points raised during the review process.

We look forward to receiving your revised manuscript.

Kind regards,

Giulio Piluso, M.Sc.

Academic Editor

PLOS ONE

Journal Requirements:

“This work was supported by a grant from the Faculty of Medicine, Public Health and Nursing, Universitas Gadjah Mada, Indonesia and 3 billion, Inc, South Korea.”

“This research is funded by Faculty of Medicine, Public Health and Nursing, Universitas Gadjah Mada, Indonesia and 3 billion, Inc, South Korea. The funders had no role in study design, data collection and analysis, decision to publish, or preparation of the manuscript.”

Additional Editor Comments (if provided):

Both reviewers, who are expert in the field, have raised some criticism that authors have to address. In my opinion, the manuscript would furtherly benefit by adding a table detailing all mutations identified in DMD patients included in the study. This table would help the reader in interpreting the impact of the variants on the different Dp isoforms.

Reviewers' comments:

Reviewer's Responses to Questions

**Comments to the Author**

1. Is the manuscript technically sound, and do the data support the conclusions?

Reviewer #1: Partly

Reviewer #2: Partly

2. Has the statistical analysis been performed appropriately and rigorously? 

Reviewer #1: Yes

Reviewer #2: N/A

3. Have the authors made all data underlying the findings in their manuscript fully available?

Reviewer #1: Yes

Reviewer #2: Yes

4. Is the manuscript presented in an intelligible fashion and written in standard English?

Reviewer #1: Yes

Reviewer #2: Yes

5. Review Comments to the Author

Reviewer #1: Duchenne muscular dystrophy (DMD) is an X-linked recessive neuromuscular disorder which causes progressive muscle weakness leading to loss of ambulation in the mid-adolescent years. Affected males generally present in the first few years of life with motor symptoms and enlarged calves. However, neurodevelopmental disorders are increasingly recognized features and can be the initial presenting symptoms. The Authors report the results of a cross-sectional study aiming to characterize cognitive and neurodevelopmental profiles in a cohort of 48 patients with molecular diagnosis of DMD, aged between 5 and 18 years. Cognitive function was measured by intelligence quotient testing using the Wechsler Intelligence Scale for Children, and adaptive function tests with Vineland Adaptive Behavior Scales. The Autism Mental Status Exam and Abbreviated Conner's Rating Scale were used to screen autism spectrum disorder (ASD) and attention deficit and hyperactivity disorder (ADHD), respectively.

They found that patients with DMD have in general an IQ score lower compared to their peers and present with intellectual disability in about 20% of cases. No case of ASD, and only two boys with ADHD are reported. When trying a genotype-phenotype correlation, they conclude that DMD patients with mutation affecting Dp71 isoform of dystrophin are at higher risk to develop intellectual disability, but not ASD or ADHD.

The study is not original, as the knowledge that mutations disrupting the dystrophin brain isoforms Dp140 and Dp71 are more frequently associated with lower IQ scores in DMD patients, is not new. However, some results, such as the absence of patients with ASD and the low rate of patients with ADHD are interesting as they differ from what has been reported so far.

Critical points

- In tables there is no mention of the number of patients included in each dystrophin disruption group (Dp427, Dp140 e Dp71);

- Furthermore, in the tables it should be better clarified the difference between which groups the significance is referred;

- It would also be interesting to subdivide the patients within the same group according to the type of mutation - deletions vs duplications - and evaluate, if any, differences in these subgroups.

- In my opinion, the absence of patients with ASD and the low rate of patients with ADHD should be underlined and widely discussed as they differ markedly from what has been reported so far.

Reviewer #2: The article is well done and the topic is very interesting.

Anyway, data presented, even of interesting, appear partially in line with what available in literature. This could be due to a too small cohort or due to the application of evaluation methods not as sensitive as needed to analyze diseases object of investigation.

As said by the Author themselves, infact, the "Autism mental status exam" used by them, has a sensitivity of 81.2%.

Therefore, it could have been used a different diagnostic methods, more sensible and specific than previous

(i.e ADI-ADOS-2).

Another diagnostic method could also be used for the diagnosis of attention deficit disorder (Nepsy II in its dedicated subparts).

Moreover, since the age of the sample is very large and Duchenne muscular dystrophy is a progressive pathology, it could also be useful to differentiate the cohort according to the level of severity of symptoms (for example ambulatory and non-ambulatory).

Concerning IQ score much lower in dp71 group, this data is partially confirmed in literature but, in my opinion, this group should be enlarge to provide a wider benchmark and compare it.

For these findings, the paper need major work with a further evaluation for publication

6. PLOS authors have the option to publish the peer review history of their article (what does this mean?). If published, this will include your full peer review and any attached files.

Reviewer #1: No

Reviewer #2: No

---

## [Author Response · Author response to Decision Letter 0]

12 Jul 2022

Journal Requirements:

Thank you, we have checked and met PLOS ONE’s style requirements and file naming.

Thank you, we have provided details regarding participant consent in the Methods section and Ethics Statement field:

“Patients older than 12 years old and/or their parents or guardians (for patients <12 years old) signed a written informed consent form to be included in this study. The Institutional Review Board of the Faculty of Medicine, Public Health and Nursing, Universitas Gadjah Mada/Dr. Sardjito General Hospital approved this study (KE/FK/0894/EC/2020 and KE/FK/0180/EC/2021).”

“This work was supported by a grant from the Faculty of Medicine, Public Health and Nursing, Universitas Gadjah Mada, Indonesia and 3 billion, Inc, South Korea.”

"This research is funded by the Faculty of Medicine, Public Health and Nursing, Universitas Gadjah Mada, Indonesia and 3 billion, Inc, South Korea. The funders had no role in study design, data collection and analysis, decision to publish, or preparation of the manuscript.”

We have omitted the funding-related text from the manuscript. We agree on the statement in the funding statement section of the online submission form.

We have included amended statements in the cover letter. Thank you for your kind help.

Additional Editor Comments (if provided):

Both reviewers, who are expert in the field, have raised some criticism that authors have to address. In my opinion, the manuscript would furtherly benefit by adding a table detailing all mutations identified in DMD patients included in the study. This Table would help the reader in interpreting the impact of the variants on the different Dp isoforms.

Thank you for your suggestion. We have added a table detailing all mutations identified in DMD patients included in the study (Table 1).

Reviewers' comments:

Comments to the Author

Reviewer #1: 

Duchenne muscular dystrophy (DMD) is an X-linked recessive neuromuscular disorder which causes progressive muscle weakness leading to loss of ambulation in the mid-adolescent years. Affected males generally present in the first few years of life with motor symptoms and enlarged calves. However, neurodevelopmental disorders are increasingly recognized features and can be the initial presenting symptoms. The Authors report the results of a cross-sectional study aiming to characterize cognitive and neurodevelopmental profiles in a cohort of 48 patients with molecular diagnosis of DMD, aged between 5 and 18 years. Cognitive function was measured by intelligence quotient testing using the Wechsler Intelligence Scale for Children, and adaptive function tests with Vineland Adaptive Behavior Scales. The Autism Mental Status Exam and Abbreviated Conner's Rating Scale were used to screen autism spectrum disorder (ASD) and attention deficit and hyperactivity disorder (ADHD), respectively.

They found that patients with DMD have in general an IQ score lower compared to their peers and present with intellectual disability in about 20% of cases. No case of ASD, and only two boys with ADHD are reported. When trying a genotype-phenotype correlation, they conclude that DMD patients with mutation affecting Dp71 isoform of dystrophin are at higher risk to develop intellectual disability, but not ASD or ADHD.

The study is not original, as the knowledge that mutations disrupting the dystrophin brain isoforms Dp140 and Dp71 are more frequently associated with lower IQ scores in DMD patients, is not new. However, some results, such as the absence of patients with ASD and the low rate of patients with ADHD are interesting as they differ from what has been reported so far.

Thank you very much for these encouraging comments from this reviewer. We have emphasized our findings from previous studies in the Discussion section, particularly the absence of patients with ASD and the low rate of patients with ADHD compared to the previous result.

Critical points

- In tables there is no mention of the number of patients included in each dystrophin disruption group (Dp427, Dp140 e Dp71);

Thank you for your suggestions. We have added the number of patients included in each dystrophin disruption group in table 2 and table 3.

- Furthermore, in the tables it should be better clarified the difference between which groups the significance is referred.

We have clarified the difference between which groups the significance is referred to. We have added, "The Dp71 group significantly had the lowest cognitive function with a total IQ score of 46 ± 24.83 compared to dp427 and dp140 group (p=0.005 and 0.001, consecutively) accompanied with scores for verbal abilities of 49.75 ± 26.73 (p=0.013 and 0.005, consecutively), and for performance abilities of 45.25 ± 24.80 (p=0.001 and 0.002, consecutively)” at result section.

- It would also be interesting to subdivide the patients within the same group according to the type of mutation - deletions vs duplications - and evaluate, if any, differences in these subgroups.

We have subdivided the patient within the same group according to the type of mutation (deletion, duplication, and point mutation) and evaluated the difference in cognitive profile, the prevalence of intellectual disability, ADHD, and ASD between the group (Table 5).

- In my opinion, the absence of patients with ASD and the low rate of patients with ADHD should be underlined and widely discussed as they differ markedly from what has been reported so far.

Thank you very much for your opinion. We have underlined and added discussion about the absence of patients with ASD, and the low rate of patients with ADHD: “Correlation between types of mutation (deletion, duplication, and point mutation) and severity of symptoms (ambulatory and non-ambulatory) with cognitive profiles and prevalence of intellectual disability and ADHD in each dystrophin disruption group were also analyzed. No significant difference was found in the IQ mean and the prevalence of intellectual disability and ADHD in each type of mutation. At the same time, there is a significant correlation between mutation types and adaptive and communication functions in the Dp427 group. Previous studies also reported no significant difference in mean IQ in each type of mutation [23]. Furthermore, the significant difference in cognitive profiles, adaptive profiles, and the prevalence of intellectual disability and ADHD was not observed for the severity of symptoms in each group.”

Reviewer #2: 

The article is well done and the topic is very interesting.

Anyway, data presented, even of interesting, appear partially in line with what available in literature. This could be due to a too small cohort or due to the application of evaluation methods not as sensitive as needed to analyze diseases object of investigation.

As said by the Author themselves, infact, the "Autism mental status exam" used by them, has a sensitivity of 81.2%.

Therefore, it could have been used a different diagnostic methods, more sensible and specific than previous (i.e., ADI-ADOS-2).

Thank you very much for your kind comments. We have added “Other different diagnostic methods that are more sensible and more specific could have been used (i.e., ADI-ADOS-2)” in the Discussion section: “Other different diagnostic methods that are more sensible and more specific could have been used (i.e., ADI-ADOS-2).” 

Another diagnostic method could also be used for the diagnosis of attention deficit disorder (Nepsy II in its dedicated subparts).

We have added “Another instrument to diagnose ADHD also available, for example Nepsy II in its dedicated subparts” in the Discussion section.

Moreover, since the age of the sample is very large and Duchenne muscular dystrophy is a progressive pathology, it could also be useful to differentiate the cohort according to the level of severity of symptoms (for example ambulatory and non-ambulatory).

We added a table differentiating the cohort according to the severity of symptoms. There is no difference in IQ, ASD, and ADHD according to the severity of symptoms (Table 6).

Concerning IQ score much lower in dp71 group, this data is partially confirmed in literature but, in my opinion, this group should be enlarge to provide a wider benchmark and compare it.

As suggested by the reviewer, we have already added more patients to this study. The new Table has been inserted in the text accordingly. Unfortunately, there were no additional patients with Dp71 mutation.

For these findings, the paper need major work with a further evaluation for publication

We have added additional DMD patients for further analysis and expanded the Discussion section.

---

## [Decision Letter · Decision Letter 1]

18 Aug 2022

PONE-D-22-04415R1Dp71 and intellectual disability in Indonesian patients with Duchenne muscular dystrophyPLOS ONE

Dear Dr. Iskandar,

Thank you for submitting your manuscript to PLOS ONE. After careful consideration, we feel that it has merit but does not fully meet PLOS ONE’s publication criteria as it currently stands. Therefore, we invite you to submit a revised version of the manuscript that addresses the points raised during the review process.

We look forward to receiving your revised manuscript.

Kind regards,

Giulio Piluso, M.Sc.

Academic Editor

PLOS ONE

Journal Requirements:

Additional Editor Comments (if provided):

Concerning table 1, I believe the identified mutations should be reported according to the HGVS nomenclature. It's useful indicate the interval of exons deleted or duplicated but authors should complete table also reporting the expected phenotype DMD/BMD. Similarly, the indication "point mutation" for some patients seems very incomplete. These changes and that suggested by reviewers are mandatory.

Reviewers' comments:

Reviewer's Responses to Questions

**Comments to the Author**

1. If the authors have adequately addressed your comments raised in a previous round of review and you feel that this manuscript is now acceptable for publication, you may indicate that here to bypass the “Comments to the Author” section, enter your conflict of interest statement in the “Confidential to Editor” section, and submit your "Accept" recommendation.

Reviewer #1: All comments have been addressed

2. Is the manuscript technically sound, and do the data support the conclusions?

Reviewer #1: Yes

3. Has the statistical analysis been performed appropriately and rigorously? 

Reviewer #1: Yes

4. Have the authors made all data underlying the findings in their manuscript fully available?

Reviewer #1: Yes

5. Is the manuscript presented in an intelligible fashion and written in standard English?

Reviewer #1: Yes

6. Review Comments to the Author

Reviewer #1: The present version of the manuscript is much improved compared to the previous one. The Authors adressed all my suggestions.

I still have a few comments:

- Table 1. In the deletion group Dp427, the deletion 3-7 (patient NRR, 15 years) is listed: this is reported among the Malhotra’s exceptions to the reading frame rule. In fact, it is associated to a BMD severe phenotype.

Similarly in the deletion group Dp140, the deletions 48-49 (patient MA, 8 years) and 45-49 (patient AS, 11 years) are listed. Both these deletions are in frame and usually result in a BMD phenotype. These three patients must therefore be eliminated from the study.

- Table 2. The Measure Unit for CK and CKMB should be indicated;

- Table 3. WISC group. Specify better that the significance of “p” is between the Dp71 group and the other two groups, as I understand or not;

- Table 4. Specify better the significance for the intellectual disability to which group it refers;

- pag.15, line 17. I would say “motor” skills rather than “motoric” skills.

7. PLOS authors have the option to publish the peer review history of their article (what does this mean?). If published, this will include your full peer review and any attached files.

Reviewer #1: **Yes: **Luisa Politano

---

## [Author Response · Author response to Decision Letter 1]

3 Oct 2022

RESPONSES TO REVIEWER AND EDITOR:

The editor and reviewer comments are in italics in the material below, and our responses are regular.

Journal Requirements:

Our manuscript has not cited papers that have been retracted.

Additional Editor Comments (if provided):

Concerning table 1, I believe the identified mutations should be reported according to the HGVS nomenclature. It's useful indicate the interval of exons deleted or duplicated but authors should complete table also reporting the expected phenotype DMD/BMD. Similarly, the indication "point mutation" for some patients seems very incomplete. These changes and that suggested by reviewers are mandatory.

Thank you for your suggestions. We have already specified the mutation detail as for point mutation according to HGVS nomenclature. Moreover, we have excluded three patients with the expected BMD phenotype as per reviewer #1 suggestion, so all patients have DMD phenotype.

Reviewers' comments:

Review Comments to the Author

Reviewer #1: The present version of the manuscript is much improved compared to the previous one. The Authors adressed all my suggestions.

I still have a few comments:

- Table 1. In the deletion group Dp427, the deletion 3-7 (patient NRR, 15 years) is listed: this is reported among the Malhotra’s exceptions to the reading frame rule. In fact, it is associated to a BMD severe phenotype.

Similarly in the deletion group Dp140, the deletions 48-49 (patient MA, 8 years) and 45-49 (patient AS, 11 years) are listed. Both these deletions are in frame and usually result in a BMD phenotype. These three patients must therefore be eliminated from the study.

Thank you for your comments. We have already excluded the deletions 3-7 (patient NRR, 15 years) from the deletion group Dp427 also the deletions 48-49 (patient MA, 8 years) and 45-49 (patient AS, 11 years) from the deletion group Dp140.

- Table 2. The Measure Unit for CK and CKMB should be indicated;

Thank you for your suggestion. We have already added the measuring unit for CK (U/L) and CKMB (U/L) in Table 2.

- Table 3. WISC group. Specify better that the significance of “p” is between the Dp71 group and the other two groups, as I understand or not;

Thank you for your suggestions. We have already added the following sentences: “The Dp71 group significantly had the lowest cognitive function with total IQ score of 46.0 ± 24.8 compared to the other two groups (p=0.003) accompanied with scores for verbal abilities of 49.75 ± 26.73 (p=0.009), and for performance abilities of 45.3 ± 24.8 (p=0.002).”

- Table 4. Specify better the significance for the intellectual disability to which group it refers;

Thank you for your suggestions. We have already added the following sentences: “The incidence of intellectual disability differed significantly between the Dp71 group and the other two groups (p=0.009). All patients within the Dp71 group had intellectual disabilities.”

- pag.15, line 17. I would say “motor” skills rather than “motoric” skills.

Thank you for your suggestion. We have already replaced “motoric” skills with “motor” skills.

---

## [Decision Letter · Decision Letter 2]

11 Oct 2022

Dp71 and intellectual disability in Indonesian patients with Duchenne muscular dystrophy

PONE-D-22-04415R2

Dear Dr. Iskandar,

We’re pleased to inform you that your manuscript has been judged scientifically suitable for publication and will be formally accepted for publication once it meets all outstanding technical requirements.

Kind regards,

Giulio Piluso, M.Sc.

Academic Editor

PLOS ONE

Additional Editor Comments (optional):

Reviewers' comments:

Reviewer's Responses to Questions

**Comments to the Author**

1. If the authors have adequately addressed your comments raised in a previous round of review and you feel that this manuscript is now acceptable for publication, you may indicate that here to bypass the “Comments to the Author” section, enter your conflict of interest statement in the “Confidential to Editor” section, and submit your "Accept" recommendation.

Reviewer #1: All comments have been addressed

2. Is the manuscript technically sound, and do the data support the conclusions?

Reviewer #1: Yes

3. Has the statistical analysis been performed appropriately and rigorously? 

Reviewer #1: Yes

4. Have the authors made all data underlying the findings in their manuscript fully available?

Reviewer #1: Yes

5. Is the manuscript presented in an intelligible fashion and written in standard English?

Reviewer #1: Yes

6. Review Comments to the Author

Reviewer #1: The present revision of the manuscript clarified the points I underlined. All my comments have been accepted and the manuscript changed accordingly.

I have no further requests.

7. PLOS authors have the option to publish the peer review history of their article (what does this mean?). If published, this will include your full peer review and any attached files.

Reviewer #1: **Yes: **Luisa Politano

---

## [Editor Report · Acceptance letter]

21 Oct 2022

PONE-D-22-04415R2 

Dp71 and intellectual disability in Indonesian patients with Duchenne muscular dystrophy 

Dear Dr. Iskandar:

I'm pleased to inform you that your manuscript has been deemed suitable for publication in PLOS ONE. Congratulations! Your manuscript is now with our production department. 

Kind regards, 

on behalf of

Professor Giulio Piluso 

Academic Editor

PLOS ONE